# Effects of Microwave Pasteurization on the Quality and Shelf-Life of Low-Sodium and Intermediate-Moisture Pacific Saury (*Cololabis saira*)

**DOI:** 10.3390/foods12102000

**Published:** 2023-05-15

**Authors:** Shibin Wang, Ji Zhang, Yifen Wang, Qingcheng Zhu, Xiaodong Wang, Donglei Luan

**Affiliations:** 1College of Food Science and Technology, Shanghai Ocean University, Shanghai 201306, China; shibinwang1996@163.com (S.W.); kg3630@163.com (J.Z.); 2Biosystems Engineering Department, Auburn University, Atlanta, GA 36849, USA; wangyif@auburn.edu; 3National Engineering Research Center for Pelagic Fishery, Shanghai 201306, China; qczhu@shou.edu.cn (Q.Z.); xdwang@shou.edu.cn (X.W.)

**Keywords:** microwave pasteurization, intermediate-moisture foods, ready-to-eat food products, low-sodium foods, shelf-life

## Abstract

The objective of this study was to investigate the effects of microwave pasteurization on the quality and shelf-life of low-sodium and intermediate-moisture Pacific saury. Microwave pasteurization was used to process low-sodium (1.07% ± 0.06%) and intermediate-moisture saury (moisture content 30% ± 2%, water activity 0.810 ± 0.010) to produce high-quality ready-to-eat food stored at room temperature. Retort pasteurization with the same thermal processing level of F_90_ = 10 min was used for comparison. Results showed that microwave pasteurization had significantly (*p* < 0.001) shorter processing times (9.23 ± 0.19 min) compared with traditional retort pasteurization (17.43 ± 0.32 min). The cook value (C) and thiobarbituric acid (TBARS) content of microwave-pasteurized saury were significantly lower than that of retort-pasteurized saury (*p* < 0.05). With more microbial inactivation, microwave pasteurization brought better overall texture than retort processing. After 7 days of storage at 37 °C, the total plate count (TPC) and TBARS of microwave pasteurized saury still met the edible standard, while the TPC of retort pasteurized saury no longer did. These results showed that the combined processing of microwave pasteurization and mild drying (Aw < 0.85) could produce high-quality ready-to-eat saury products. These results indicate a new methodology for producing high-quality products stored at room temperature.

## 1. Introduction

The demands of high-quality convenient food products are increasing rapidly with the fast-paced life of modern consumers [1,2]. Drying and thermal processing are the two most widely used techniques for producing convenient ready-to-eat food products. However, the drying process is very time- and energy-consuming, while thermal processing leads to serious nutrition and sensory deterioration [3,4]. Thus, food additives are usually utilized to reduce the processing intensity, which could improve product quality and extend shelf-life. These types of ready-to-eat food products could not meet the increasing demands of customers for high-quality healthy foods. Thus, several novel processing technologies are developing to improve the nutritional and sensory quality of food products while reducing the amount of food additives.

The quality of food always decreases during any processing, and the decreasing rate depends on the processing intensity. Thus, hurdle technology has been developed in the food industry, which combines several mild processing technologies to achieve the target process instead of one single processing with high intensity [5]. Theoretically, drying can reduce the moisture content and water activity (Aw) of food, which prevents microbial growth, enzyme activity, and potential chemical reactions. As a result, dried food products are always stored at room temperature. For the traditional drying process, foods are kept in flowing hot air for several hours or even longer. This process takes a very long time and substantial energy, especially in the falling rate period of the drying process. During this period, it is hard to remove water from the interior of food products [6]. Thus, severe quality degradation occurs during this long period. In general, during thermal processing, 121.1 °C is the commonly used standard reference temperature for traditional processes to inactivate the pathogenic and spoilage bacteria. The temperature at the geometry center of food can reach 118 °C or higher, while the surface endures 121.1 °C or higher temperatures for a very long time [2,7]. This can cause severe damage to heat-sensitive ingredients.

According to the principle of hurdle technology, the combination of mild drying and thermal processing can improve product quality. Intermediate drying combined with pasteurization is a good alternative to processing instead of complete drying or thermal sterilization [8,9]. This combined process would greatly shorten the drying time and reduce the thermal processing intensity.

However, traditional thermal pasteurization still requires a long processing time due to the low heat transfer rate within intermediate-moisture foods. To overcome this disadvantage, microwave thermal pasteurization can be combined to replace traditional thermal pasteurization. Microwave heating is a novel thermal processing technology with a fast heating rate [10,11,12]. Previous studies showed that a rapid heating rate reduced the thermal resistance of microorganisms, enabling easier inactivation [13]. Furthermore, microwaves can bring extra microbial inactivation due to the nonthermal effects of microwave fields [14,15], representing another hurdle factor. These characteristics of microwave heating are beneficial to the quality improvement of food while ensuring its shelf-life [16,17]. Therefore, intermediate-moisture foods can be processed by microwave pasteurization as a novel technique to produce high-quality, ready-to-eat foods.

Pacific saury is rich in unsaturated fatty acids including EPA and DHA, which is beneficial to human health. Canned saury is the most common processing of saury on the market. However, polyunsaturated fatty acids within saury are easily oxidized during high-temperature long-time processing [18,19]. Furthermore, the texture of saury is seriously damaged. Additionally, excessive salt and food additives are usually applied. The canned saury on the current market cannot meet the demands of modern consumers for high-quality, low-sodium products. Therefore, novel combined processing technology is necessary to improve product quality while ensuring shelf-life.

The objectives of this study were to investigate the effects of microwave pasteurization on the quality of low-sodium and intermediate-moisture saury and its shelf-life at room temperature. After 7 days of storage at 37 °C, the microbial and quality indicators still met the edible standard; hence, the saury can theoretically be stored at room temperature). These results can provide theoretical fundamentals for the production of healthy and convenient food products.

## 2. Materials and Methods

### 2.1. Sample and Reagents Preparation

Pacific saury (*Cololabis saira*) was purchased from the local wholesale aquatic products market (Shanghai, China) and stored at −18 °C. Before processing, each saury was thawed at 0–4 °C for 12 h. After removing the head and guts, saury was weighed and numbered for the subsequent drying process. Thiobarbituric acid (AR, ≥98%) was purchased from Macklin Biochemical Co., Ltd. (Shanghai, China). Petroleum ether 30–60 °C (AR) was purchased from Collins Chemical Co., Ltd. (Shanghai, China).

### 2.2. Moderate Drying and Sousing Processing

The saury was dried at 55 °C in a vacuum drying oven (CIMO, Shanghai, China) with a pressure of 0.1 MPa. When the moisture content of the saury decreased to 30%, it was taken out for 1.5 h soused processing within commercial liquid synthesis flavoring (Niulege, Chengdu, China). The ingredients of the synthesis flavoring included ginger juice, garlic powder, cooking wine, salt, and polyphosphates. The ratio of saury and liquid flavoring was 3:1 by weight. After sousing, the saury continued to vacuum-dry at 55 °C for 1 h; the fish lost water reaching 30% ± 2% moisture. Dried saury was cut into pieces with sizes of 50 mm × 20 mm × 14 mm; then, each piece was vacuumed and packaged for the subsequent thermal pasteurization process.

### 2.3. Thermal Pasteurization Process

#### 2.3.1. Retort Pasteurization Process

Traditional thermal pasteurization was carried out using an HX-320 retort (Systec, Heinsdorfergrund, Germany) with the heating medium of steam as control. The temperature for pasteurization was set at 93 °C and the saury was held for 8.5 min when the inner temperature of the retort cavity reached 93 °C. Saury was taken out at the end of holding processing and placed into cold water for cooling. During thermal processing, a mobile metallic temperature sensor (PICOVACQ/1TC, TMIORION, France) was used to record the time–temperature profile at the cold spot location of the saury. All treatments were repeated three times.

#### 2.3.2. Microwave Pasteurization Process

Microwave pasteurization was performed using an 896 MHz single-mode pilot-scale microwave processing system installed at Shanghai Ocean University (Shanghai, China). A structural sketch of the microwave processing system is shown in Figure 1. The system consisted of four sections: loading, microwave heating, holding, and cooling [15]. The samples of vacuumed packaged saury were transported to the four sections in succession by a carrier. Within the microwave radiation and holding section, samples were immersed in a waterbed at 93 °C. During microwave heating, the microwave net power was 7 kW. With a moving speed of 2.2 cm/s, each sample received a microwave radiation time of 2 min. At the end of the microwave heating section, the temperature at the cold spot of saury reached about 89–90 °C. Following 4 min processing in 93 °C hot water, the thermal processing level of each saury sample achieved F_90_ = 10 min. A mobile metallic temperature sensor was used to record the time–temperature profile at the cold spot location of the saury. The cold spot location was confirmed by the chemical marker method combined with a computer vision system [20,21]. All treatments were repeated three times.

### 2.4. Thermal Processing Level and Cook Value

The thermal processing level (F) is an equivalent of the thermal processing time at a constant reference temperature, which is used to evaluate the microbial lethal effect as the conducted thermal processing. It is calculated using the following equation based on the time–temperature profiles of the conducted thermal processing. Normally, fish products can achieve the pasteurization effect with the thermal processing level of F_90_ = 10 min [22].
$$F=\int_{0}^{t}10^{\frac{T_{\left(t\right)}-T_{\mathrm{ref}}}{z}}dt,$$
where t represents time (min), T_(t)_ is the temperature at the cold spot at time t during processing (°C), T_ref_ is the reference temperature, and z is the z-value of the target microorganism in the products; in this experiment, it was calculated by default at 10 °C [2].

The cook value (C) is used to evaluate the cumulative thermal effect of time and temperature on food quality (nutrients, texture, etc.). It can reflect the loss of food quality caused by thermal processing at the reference temperature.
$$C=\int_{0}^{t}10^{\frac{T_{\left(t\right)}-100}{z_{n}}}dt,$$
where t represents time (min), T_(t)_ is the same as used in calculating F, $z_{n}$ is the z-value for food nutrients (sensitive to temperature changes; generally taken as 33.1 °C), and 100 represents a heat treatment equivalent to 100 °C.

### 2.5. Quality Evaluation

#### 2.5.1. Moisture Content and Water Activity (Aw)

The moisture content of saury was measured using the direct drying method. The HD-3B intelligent Aw meter (Huake, Wuxi, China) was used to measure the Aw of raw and processed saury samples. Saturated salt water was used to calibrate the Aw meter before testing. The saury sample was minced and placed in the sample box. Then, it was transferred to the sample pool for testing. During testing, the sample temperature was controlled at 25 °C.

#### 2.5.2. Sodium Content

Sodium content was measured by inductively coupled plasma mass spectrometry (ICP-MS), according to a previously published method with minor modification [23]. Briefly, a 0.5 g sample of saury was weighed (accurate to 0.001 g) and put into the PTFE digestion tank. Next, 5 mL of nitric acid was added; the saury was predigested overnight, and then digested by a microwave digestion system (CEM company, Boston, MA, USA). The digestion solution was diluted 100 times. Finally, Na^+^ was measured using an inductively coupled plasma mass spectrometer (ICAP Q inductively coupled plasma mass spectrometer, Thermo Fisher Scientific Inc., Waltham, MA, USA). All measurements were performed in three replications.

#### 2.5.3. Texture Profile Analysis (TPA)

The texture of the sample was analyzed using a TA-XT plus texture analyzer (Stable Micro Systems Ltd., Godalming, UK) equipped with a cylindrical flat-bottom aluminum probe; the diameter of the probe was 6 mm [24,25]. Samples were prepared at room temperature (20–25 °C). The measuring parameters were set as follows: pre-test speed, test speed, and return speed of 1.0 mm/s, 1.0 mm/s, and 5.0 mm/s, respectively. In addition, the compression was 30%, the holding time was 5 s, and the trigger force was 0.05 N. The values of hardness, springiness, gumminess, cohesiveness, and chewiness were calculated using the texture expert software, version 1.22 (Stable Micro Systems Ltd., Godalming, UK). All measurements were performed in six replications. 

#### 2.5.4. Lipid Oxidation

Lipid oxidation is a complex chemical reaction process. Firstly, in lipid oxidation to form peroxides, the peroxide value (POV) is used to measure the primary product of lipid oxidation. However, peroxides are unstable and can be further oxidized to produce aldehydes, ketones, and other compounds. Malondialdehyde (MDA) is the most representative secondary oxidation product, which is usually measured as TBARS [26]. The steps for measuring the POV were as follows: firstly, the crude fat of saury fish was extracted using the Soxhlet extraction method. Secondly, a saturated potassium iodide solution was added to the oil, which was finally titrated with a sodium thiosulfate standard solution until the blue color disappeared. The content of POV was calculated on the basis of the reagents consumed during the titration process. The TBARS was measured according to the following method: briefly, 5.0 g of homogenized fish was put into a beaker, and 25 mL of 7.5% trichloroacetic acid solution was added, thoroughly stirred, and filtered after shaking for 30 min. Then, 5 mL of the above solution was removed into the test tube, and 5 mL of 0.02 mol/L thiobarbituric acid solution was added. The mixed reagent was heated in 90 °C water for 40 min. After heating, the test tube was removed and cooled at room temperature. Finally, the absorbance of the mixed reagent at 532 nm was measured using a spectrophotometer. The TBARS value was expressed as mg (MDA)/kg saury [27]. The saury was stored at 37 °C for 7 days, and the POV and TBARS were determined using the same method as above.

### 2.6. Total Plate Count (TPC)

The total plate count was measured using the plate colony counting method. The medium for counting total microorganisms was plate counting agar (PCA). Minced pasteurized saury was put into aseptic physiological saline, and the mixture was shaken in an aseptic homogenization bag using a stomacher blender (Lab-blender 400, Licheng company, Ningbo, China) for 60 s, before being diluted successively to the appropriate concentration for counting. The counted plates were cultured at 37 °C for 48 h. Different samples were analyzed in duplicate, and the results were expressed as the log colony-forming unit per gram (CFU/g) of a sample. The saury was stored at 37 °C for 7 days, and the TPC was determined using the same method as above.

### 2.7. Statistical Analysis

The experimental data were analyzed using SPSS 25 software (SPSS Inc., Chicago, IL, USA). The results of repeated experiments are shown as the mean and standard deviation. The significance (*p* < 0.05) was analyzed by one-way analysis of variance and Duncan’s multiple range test.

## 3. Results and Discussion

### 3.1. Moisture Sorption Isotherm (MSI) and Moisture Content Curve with Drying Time of Saury

The moisture sorption isotherm shows the corresponding relationship between moisture content (wet basis, water/total weight) and Aw of saury [28]. Figure 2A clearly demonstrates that the Aw of saury declined with the decreasing moisture content. When the moisture content decreased from 60% to 20.18%, the Aw declined from 0.920 to 0.855. The moisture content of intermediate moisture foods was 20–50%, and the Aw was 0.7–0.85 [29]. Therefore, when the moisture content of saury reached 20.18%, it was close to the critical point of intermediate-moisture foods and dried products. Figure 2B shows the moisture content curve with drying time of saury. When the moisture content decreased from 60% to 20%, the drying time consumed was close to 10 h. Long-time drying is accompanied by a significant amount of energy consumption, and food quality may deteriorate during this process. According to Figure 2A, when the moisture content was 30.48%, the Aw of saury was 0.868, which was approximated to 0.850. Meanwhile, as shown in Figure 2B, when the moisture content of saury was 30%, the required drying time was only 5 h. To save energy while maintaining the good appearance and palatability of saury, the moisture content of saury was controlled at about 30%. In order to extend the shelf-life of final products, the Aw should be as low as possible. Therefore, it is necessary to take some effective methods to further reduce the Aw while maintaining a moderate moisture content.

### 3.2. Water Activity Changing of Saury with Different Treatments

The Aw changes of saury with different treatments are shown in Figure 3. At the beginning, the Aw of raw saury was 0.920. After drying treatment until the moisture content of saury was reduced to 30%, the Aw was 0.868. Sousing has been shown to be an effective method of reducing Aw [30]. This is because sodium salt exists in the form of ions when it is dissolved in water and interacts with water molecules to form a dipole ion structure, which restricts the mobility of water molecules, thereby reducing Aw [30,31]. After the sousing treatment, the Aw of saury was reduced to 0.825. In order to obtain a final moisture content of 30%, a secondary drying treatment was performed to remove the water gained during the souring process. Through the synergistic effect of drying, sousing, and second drying, the Aw of saury was significantly reduced to 0.810. An intermediate-moisture saury with a moisture content (30% ± 2%) and an Aw (0.810 ± 0.010) was obtained.

### 3.3. Sodium Content

Currently, the salt intake in most countries far exceeds the intake standard of 5 g salt/day (2 g sodium/day) recommended by the World Health Organization [32,33]. Hypertension, cardiovascular disease, kidney disease, and other diseases induced by excessive sodium intake have become social problems to be solved urgently. Therefore, sodium content reduction has become a health consensus. ICP-MS can quantitatively analyze sodium in the food according to the number of ions with a specific mass-to-charge ratio of sodium element [34]. According to the test results, the sodium content of intermediate moisture saury was 1.07 ± 0.06 g/100 g. The sodium content of microwave- and retort-pasteurized saury was 1.21 ± 0.08 g/100 g and 1.28 ± 0.05 g/100 g, respectively. The sodium content of two pasteurized saury was lower than 2%. Therefore, both pasteurized saury products achieved low sodium addition [30,35].

### 3.4. Time–Temperature Profiles

The time–temperature profiles of microwave and retort pasteurization of intermediate-moisture saury are shown in Figure 4. Obviously, microwave pasteurization had a significantly (*p* < 0.001) shorter total processing time (9.23 ± 0.19 min) than retort pasteurization (17.43 ± 0.32 min), indicating that microwave pasteurization had a much faster heating rate. The rapid heating rate has been reported to be beneficial for the inactivation of microorganisms [13,36]. This is because the rapid heating rate reduces the thermal resistance of microorganisms, which are inactivated without reacting to heat. In addition, the rapid heating rate shortened the heating time, not only improving production efficiency but also saving energy [37,38]. The F value and C value for microwave and retort pasteurization were calculated on the basis of the time–temperature profiles, as shown in Table 1. The thermal processing level (F_90_) of these two pasteurization groups was almost identical (microwave pasteurization F_90_ = 10.56 min, retort pasteurization F_90_ = 10.51 min). Compared with retort pasteurization, the C value of microwave pasteurization was significantly lower (*p* < 0.001). Accordingly, microwave heating could adequately sustain the quality of saury. Therefore, microwave pasteurization may bring a longer shelf-life while maintaining good quality.

### 3.5. Texture Profile Analysis (TPA)

Texture is one of the most important quality indicators of meat products, including hardness, springiness, chewiness, cohesiveness, and gumminess. The texture results of saury after different processing are listed in Table 2. Compared with the raw sample, the hardness value of intermediate-moisture saury increased significantly (*p* < 0.05). This is because, during the drying process, saury dehydration caused muscle fibers to shrink and the tissue structure became tighter, leading to an increase in hardness value [39]. Meanwhile, the drying temperature (55 °C) caused myosin denaturation, which also increased the hardness. The hardness value was further increased for both microwave- and retort-pasteurized saury. Moreover, the hardness value of the retort-pasteurized saury was significantly higher (*p* < 0.05) than that of the microwave-pasteurized saury. This may have been due to the complete denaturation of actin and severe dehydration and shrinkage of tissue structure caused by long-term thermal processing [40,41]. Therefore, microwave-pasteurized saury reduced the degree of damage to the tissue structure, which brought better quality. Furthermore, with the extension of thermal processing time, microwaves could kill more microorganisms. This increased the hardness value while extending the shelf-life and bringing better sensory acceptance to consumers. The springiness and chewiness values of pasteurized saury were significantly (*p* < 0.05) higher than those of the raw sample and intermediate-moisture saury, but there were no significant differences between the microwave- and retort-pasteurized saury (*p* > 0.05). Similarly, in terms of cohesiveness and gumminess, there were no significant differences between the microwave- and retort-pasteurized saury. Overall, microwave-pasteurized saury products had better texture characteristics.

### 3.6. Analysis of Lipid Oxidation

The peroxide value (POV) and thiobarbituric acid reactive substances (TBARS) are indicators to evaluate the primary and secondary stages of lipid oxidation, respectively. The lipid oxidation of the saury with different processing methods is shown in Figure 5. The POV of intermediate-moisture saury was significantly higher than that of the raw sample (*p* < 0.05), indicating that lipid oxidation is inevitable during vacuum-drying. The POV of microwave- and retort-pasteurized saury further increased significantly (*p* < 0.05), but there were no significant differences between the microwave- and retort-pasteurized saury (*p* > 0.05). Moreover, the TBARS value of the retort-pasteurized saury was significantly higher than that of the microwave-pasteurized saury (*p* < 0.05). Generally, temperature and thermal processing time are two factors that affect lipid oxidation [26]. At the same temperature, traditional retort pasteurization has a long thermal processing time. Theoretically, the POV and TBARS value of traditional pasteurized saury are significantly higher (*p* < 0.05) than those of microwave-pasteurized saury. However, lipid oxidation is a complex process, and excessive oxidation would cause a primary oxidation product (POV) transition to the secondary product malondialdehyde (MDA), which is measured by TBARS content [27,42]. As a result, there was no significant difference in POV between the two groups of pasteurized saury (A). The TBARS value of the microwave pasteurization group was significantly lower (*p* < 0.05) than that of the retort pasteurization group (B). Microwave thermal pasteurization makes full use of the advantages of rapid heating rate and short-term processing to minimize the oxidation of lipid. This protection of nutrients is crucial in the production of high-quality ready-to-eat foods.

### 3.7. Total Plate Count (TPC)

Figure 6 shows the TPC values for different types of processed saury, with intermediate-moisture saury at 3.42 log CFU/g, retort-pasteurized saury at 1.46 log CFU/g, and no detectable microorganisms in the microwave-pasteurized saury. The rapid heating rate of microwaves could reduce the heat resistance of microorganisms and result in higher inactivation rates. Additionally, microwave treatment can have nonthermal effects [14,15], which further contribute to microbial inactivation. To evaluate the effect of microwave pasteurization on extending shelf-life, samples of the different processed saury were stored in a thermostatic constant-moisture incubator at 37 °C. By the seventh day, the TPC of intermediate-moisture saury was 7.97 log CFU/g, and that of retort-pasteurized saury was 5.17 log CFU/g, exceeding the national limit standard of 5 × 10^4^ CFU/g [43]. In contrast, the TPC of microwave-pasteurized intermediate moisture saury was 3.36 log CFU/g, which was within the national limit standard. Thus, from a microbial perspective, microwave-pasteurized saury products have a longer shelf-life at room temperature, all other factors being equal.

### 3.8. Lipid Oxidation on the Seventh Day of Storage

While the total plate count (TPC) is an important indicator for assessing food shelf-life, other quality indicators may also be relevant. Given the high lipid content of saury, lipid oxidation is considered a key factor influencing its shelf-life. As indicated in Table 3, after 7 days of accelerated storage at 37 °C, the peroxide value (POV) of retort-pasteurized saury was significantly higher (*p* < 0.05) than that of microwave-pasteurized saury. However, both groups remained below the upper limit of 0.6 g/100 g [44]. The thiobarbituric acid reactive substances (TBARS) content exhibited a similar trend, with neither group exceeding the 2.0 mg/kg limit for human consumption [45]. Notably, the microwave-pasteurized saury showed lower levels of POV and TBARS, possibly due to the shorter heating time, which reduced the degree of lipid oxidation [46]. This suggests that microwave-pasteurized saury may have a greater tolerance for lipid oxidation during storage, thereby extending its shelf-life. Combining these findings with the results of TPC testing, it appears that microwave-pasteurized saury has a longer shelf-life than its retort-pasteurized counterpart.

## 4. Conclusions

To produce high-quality, ready-to-eat saury, a combination of microwave pasteurization and mild drying was used, which reduced the water activity of the saury to about 0.810 through moderate drying and sousing. The intermediate-moisture saury was subjected to microwave and traditional retort pasteurization to achieve a thermal processing level of F_90_ = 10 min. The results showed that, compared to retort pasteurization, microwave pasteurization reduced the processing time by 47% and decreased the cooking value by 42%. Microwave-pasteurized intermediate-moisture saury had superior texture and lower lipid oxidation. Furthermore, microwave pasteurization inactivated a greater number of microorganisms in the intermediate-moisture saury. After 7 days of storage at 37 °C, the TPC and TBARS values of microwave-pasteurized saury were within the acceptable range for ready-to-eat foods. However, the TPC of traditionally retort-pasteurized saury exceeded the standard for ready-to-eat foods. This combined processing method of microwave pasteurization and mild drying can produce high-quality, low-sodium saury products that can be stored at room temperature, making them suitable for producing new ready-to-eat food products.

## Figures and Tables

**Figure 1 foods-12-02000-f001:**
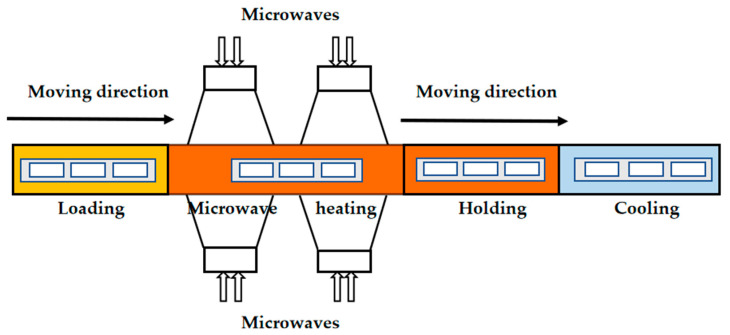
The 896 MHz microwave processing system at Shanghai Ocean University.

**Figure 2 foods-12-02000-f002:**
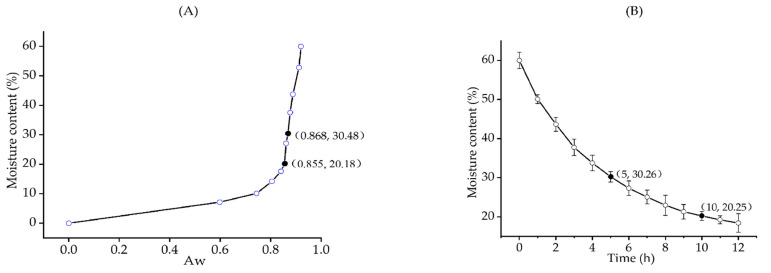
Moisture sorption isotherm of saury (**A**); moisture content curve with drying time of saury (**B**).

**Figure 3 foods-12-02000-f003:**
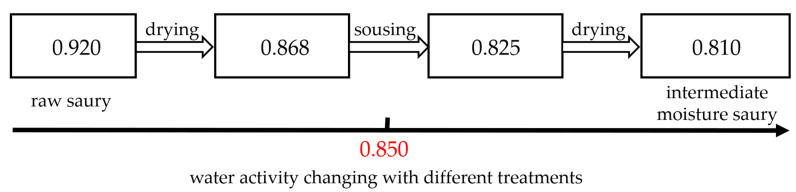
Water activity changes in saury with different treatments. A value of 0.850 is the critical point for Aw to meet the requirements of intermediate moisture foods.

**Figure 4 foods-12-02000-f004:**
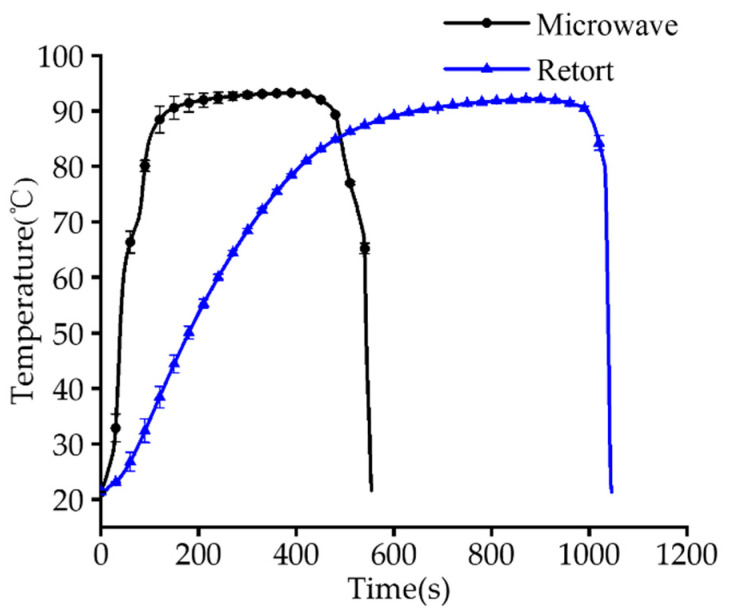
Time–temperature profiles of microwave and retort pasteurization.

**Figure 5 foods-12-02000-f005:**
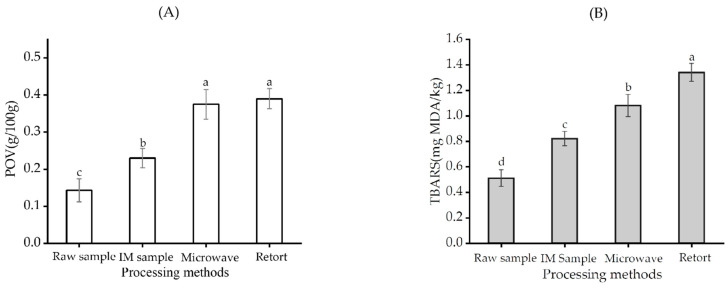
POV content of saury with different processing methods (**A**); TBARS content of saury with different processing methods (**B**). IM sample refers to the intermediate-moisture sample. Different lowercase letters in the same column indicate a significant difference (*p* < 0.05).

**Figure 6 foods-12-02000-f006:**
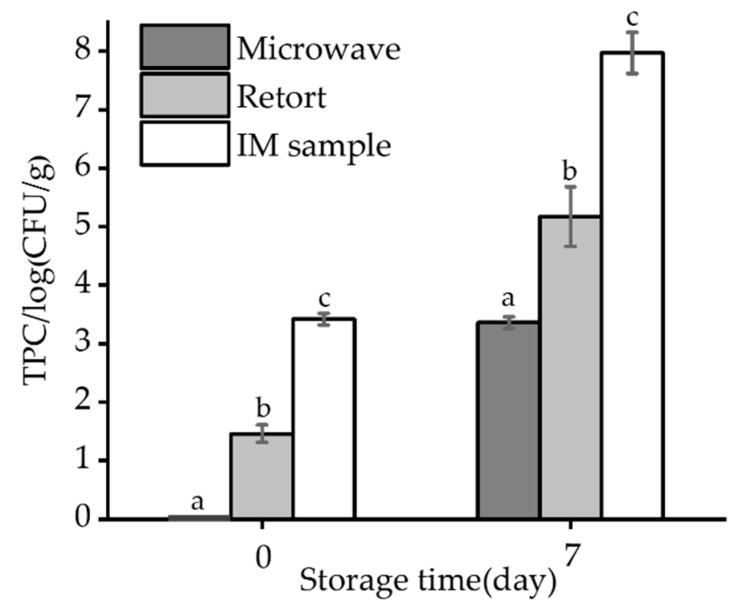
TPC of IM, retort-pasteurized, and microwave-pasteurized saury on days 0 and 7 of storage. IM sample refers to the intermediate-moisture sample. On the same day, different lowercase letters indicate a significant difference (*p* < 0.05).

**Table 1 foods-12-02000-t001:** Comparison of F value, C value, and TPT of two pasteurization processing methods.

Processing Methods	F_90_ (min)	C (min)	TPT (min)
Microwave	10.56 ± 0.23 ^a^	10.02 ± 0.24 ^a^	9.23 ± 0.19 ^a^
Retort	10.51 ± 0.18 ^a^	17.13 ± 0.42 ^b^	17.43 ± 0.32 ^b^
*p*-value	0.782	<0.001	<0.001

Note: The results are the mean ± standard deviation. Different lowercase letters in the same column indicate a significant difference; *p* > 0.05 indicates no significant difference, *p* < 0.001 indicates a highly significant difference. TPT represents total processing time.

**Table 2 foods-12-02000-t002:** Texture profile analysis of different processing methods.

Processing Methods	Texture Profile				
	Hardness (N)	Springiness	Chewiness (mJ)	Cohesiveness	Gumminess (N)
Raw sample	3.00 ± 0.38 ^d^	0.72 ± 0.02 ^b^	1.17 ± 0.07 ^c^	0.54 ± 0.03 ^b^	1.62 ± 0.12 ^b^
IM sample	11.63 ± 0.60 ^c^	0.75 ± 0.01 ^b^	5.58 ± 0.40 ^b^	0.64 ± 0.04 ^a^	7.44 ± 0.48 ^a^
Microwave	14.18 ± 0.91 ^b^	0.91 ± 0.02 ^a^	6.84 ± 1.13 ^ab^	0.53 ± 0.05 ^b^	7.52 ± 1.12 ^a^
Retort	16.31 ± 0.39 ^a^	0.90 ± 0.01 ^a^	7.78 ± 0.63 ^a^	0.53 ± 0.05 ^b^	8.64 ± 0.72 ^a^

Note: IM sample refers to the intermediate-moisture sample. The results are the mean ± standard deviation. Different lowercase letters in the same column indicate a significant difference (*p* < 0.05).

**Table 3 foods-12-02000-t003:** The POV and TBARS content of pasteurized saury stored at 37 °C for 7 days.

Processing Methods	Lipid Oxidation Indicators	
	POV (g/100 g)	TBARS (mg MDA/kg)
Microwave	0.486 ± 0.011 ^b^	1.49 ± 0.092 ^b^
Retort	0.523 ± 0.038 ^a^	1.70 ± 0.080 ^a^
*p*-value	0.005	0.039

Note: The results are the mean ± standard deviation. Different lowercase letters in the same column indicate a significant difference (*p* < 0.05).

## Data Availability

Data is contained within the article.

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
