# Peer review of "Effects of Microwave Pasteurization on the Quality and Shelf-Life of Low-Sodium and Intermediate-Moisture Pacific Saury (Cololabis saira)"

_foods, 2023, doi:10.3390/foods12102000_

Round 1

Reviewer 1 Report

The purpose of the study was to determine the “Effects of microwave pasteurization on the quality and shelf  life of low sodium and intermediate moisture Pacific saury (Cololabis saira)”.  The introduction section provides only general information.  The manuscript is rather weak scientifically. The manuscript needs to be revised taking into account the following points:

Ø  Line 52-57: “For thermal processing, food products are required a high temperature (121.1 °C or higher) sterilization process  to inactivate the pathogenic and spoiled bacteria. The temperature at the geometry center of food would reach 118 °C or higher, while the surface has been enduring 121.1 °C  or higher temperature for a very long time [2,7].”

These expressions are controversial. In thermal processing, the reference temperature is an important factor for the calculation of F value. In addition, the combination of time and temperature is more effective. It is not necessary to reach 121.1 °C. In other words, the important thing is to achieve the lethality obtained at this temperature. For example, the target F value can also be achieved at 110 °C. These expressions (Lines 52-57) should be reconsidered.

Ø  How many times was the treatment repeated?

Ø  The experimental plan should be clearly stated. Randomized complete block design or completely randomized design?

Ø   Why was pH not analyzed? In my opinion, the pH value is important for this study.

Ø  SD has not been given for some averages (Line 270-272). Are these analyses done once?

Ø  Line 393: What is the standard level?

Ø  The result of TBARS and texture profile analysis are not adequately discussed. They should be revised.

Reviewer 2 Report

The information described in the manuscript is interesting and novel, however, work is required on the following observations:

Line 89: add a materials and reagents section, including reagents used and commercial origin

Line 90: use italic text format for scientific names

Line 95: add information about the equipment used (model, brand, country)

Line 98: if possible, add the brand of the ingredients used

Line 99: …was 3:1

Line 100: use 1 h instead 1 hour

Line 101: insert space (mean ± SD)

Line 105: 2.3.1.

Line 138: use Figure 1. instead of Fig 1.

Line 139: 2.4. 

Line 160: 2.5. 

Line 161: 2.5.1.

Line 168: 2.5.2.

Line 177: 2.5.3.

Line 187: 2.5.4.

Line 192: add the meaning of TBARS

Line 198: use mL instead of ml

Line 199: remove abbreviation (TCA), because it is not used on a second occasion in the document

Line 198,200: use mL instead of ml

Line 201: remove abbreviation (TBA), because it is not used on a second occasion in the document. Add the full name of the reagent

Line 204: include information regarding the equation obtained in the calibration curve that was used for the quantification of MDA

Line 210: add information about the equipment used (model, brand, country)

Line 217: insert spaces (P < 0.05)

Line 219: 3.1. Moisture

Line 233: use 5 h instead 5 hour

Line 240: use Figure 2. instead of Fig 2.

Line 242: 3.2. Water

Line 253: insert space (mean ± SD)

Line 257: use Figure 3. instead of Fig 3.

Line 261: 3.3. Sodium

Line 262: insert space 5 g

Line 272: delete space… [30,35]

Line 273: 3.4. Time-

Line 276,277: insert space (mean ± SD)

Line 293: use Figure 4. instead of Fig 4.

Line 298: Regarding the information included in this table, was a one-way analysis of variance used to compare two treatments? Did you mean a t-test?

Line 298: It is not necessary to put the literals, with the value of P they are inferred regarding the significant differences between two mean values

Line 302: 3.5. Texture

Line 323: In the information contained in this table, use only two decimal places after the point. Also, insert space (mean ± SD)

Line 323: in a row at the bottom of the table, it is necessary to include the P value

Line 326: 3.6. Analysis

Line 326: indicate in some part of the discussion the maximum allowed value of TBARS for the product to be considered unfit for consumption

Line 327: thiobarbituric acid reactive substances

Line 335: insert spaces (P < 0.05)

Line 340: In the figures it must be indicated to which each test corresponds with (A) or (B) as in figure 2

Line 340: insert space … TBARS (mg…

Line 341: use Figure 5. instead of Fig 5.

Line 344: 3.7. Total

Line 350: delete space… [14,15]

Line 352: insert space… 37 °C

Line 363: 3.8. Lipid

Line 367: insert space… 37 °C

Line 409: it is necessary to use the correct format to cite the references, review the Microsoft word template included in the authors guide section of the journal

Reviewer 3 Report

The paper is focused on the evaluation of the microwave treatment vs conventional thermal treatment on several parameters of a type of low moisture content of fish. The paper is well-written but in my opinion, the obtained results are not new and quite predictable. My main concern is that although the differences found are attributed to microwave heating, the authors shorten this treatment not only because of the application of microwaves but also because in this treatment the product is immersed in hot water at 93 °C, which is not happening in the conventional pasteurization, extending the heating periods and therefore negatively affecting certain parameters like texture because of purge loss. I have some more comments: 

Line 54: spoilage bacteria.

Line 101: Could you revise if the fish lost water reaching 30 % of moisture or if the water lost was 30 %?  In 1 h it was possible to reduce the moisture to 30 % ? The flavoring should include the concentrations of the different ingredients. 

Line 102: Include the composition of the packaging film and the OTR

Line 106: why did the authors use sterilization equipment if the temperature was below 100 °C?

Line 105 and 113: these treatments were applied the same day after the drying process? 

Line 120: It seems that the microwave process is applied to heat the water, and the heat is transferred to the fish. According to the abstract, it seems that microwaves are directly applied to fish. Please, clarify this point. The fish has very low moisture which could affect the heating of the fish by microwave. 

Line 125: Why the authors considered this F? F90=10 min consider including the reference of this parameter. 

Line 150: 10 °C is the default z for any microorganism?

Line 217: the fish was not stored to evaluate the TBARS and microbial counts evolution?

Line 240: According to Figure 2 B, it takes more than 4 h to reduce moisture from 60 % to 30 %, previously the authors indicate only 1 h. It seems rare. Or Figure 2 corresponds to fish without brine treatment?

Line 269: The sodium contents should include standard deviations. Did the treatment significantly affect the sodium content?

Line 277-293: The main differences between treatments could not be due to the fact that in the microwave treatment, the fish is directly immersed in hot water at 93 °C,  more that the application of microwave treatment? The time was shorter not only for the microwave processing but also for the immersion in hot water, which not happens in the pasteurization process.

Line 298:What is TPT? And what does it mean?

Line 314:The thermal treatments did not produce purge loss in the packages? Could this be the reason for the texture results?

Line 345: If the samples were stored for 7 days, this fact should be explained in the material and methods section.

Line 356: Would the authors recommend storing the product at room temperature? It seems that there is a microbial risk.

Line 363: This section should be included in the 3.6 section. 

The paper is correctly written. Grammar and style are adequate.

Round 2

Reviewer 1 Report

No further comments.

Author Response

Thank you.

Reviewer 3 Report

The authors have addressed most of the modifications suggested. I recommend considering the values of texture in N, not in g. 

The quality of the English language is ok

Author Response

Thank you for your review, "the values of texture in N, not in g" has been revised.
